# META-MEMORY FOR LARGE LANGUAGE MODELS

## ABSTRACT

Integrating memory components into large language models (LLMs) can improve the generation quality for long-term conversations. However, existing memory management methods largely overlook the cognition and regulation of the memory process, lacking the capability to dynamically manage and utilize memory on demand. To address this challenge, this paper approaches Meta-Memory for Memory Management ($\mathbf{M}^4$), a novel paradigm that equips LLMs with the ability for self-monitoring and self-reflective memory management. In long-term conversations, where dialogue history accumulates continuously, the meta-memory capability of $\mathbf{M}^4$ enables LLMs to autonomously 1) identify what knowledge needs to be memorized; 2) determine how to construct and store memory; 3) monitor the correctness and validity of the acquired information; and 4) decide when to learn more and how to retrieve information to refine their responses. Experimental results on two long-term conversation datasets and one long-term question-answering dataset demonstrate that our $\mathbf{M}^4$ significantly enhances the memory management capacity of LLMs in long-term information learning, achieving more efficient storage and higher-quality response generation.

## 1 INTRODUCTION

Large language models (LLMs) have achieved breakthrough progress in a series of natural language processing (NLP) tasks (Zhao et al., 2023), including open-domain conversation. Despite the remarkable progress made, existing LLMs still have limitations in effectively utilizing long-term memory to generate responses when facing long-term or multi-session conversations due to the complexity and excessive length of the historical conversations (Du et al., 2025).

To address this limitation, a series of studies (Xu et al., 2022b;a; Bae et al., 2022; Lu et al., 2023; Jang et al., 2023; Zhang et al., 2023; Du et al., 2024; Zhong et al., 2024; Chen et al., 2025; Li et al., 2025; Ong et al., 2025; Wang et al., 2025) have focused on the management of long-term historical information as memory to generate higher-quality responses for LLM in long-term conversations. However, existing approaches typically summarize historical dialogues into coarse-grained memory representations and employ them in an undifferentiated manner during query processing. As a result, these methods lack the ability to dynamically determine which types of memory to access or how much memory information to utilize when generating responses.

Therefore, inspired by theories of Metacognition (Dunlosky & Metcalfe, 2008) and Metamemory (Flavell & Wellman, 1975; Nelson, 1990) from cognitive psychology, we introduce the concept of meta-memory into memory management for LLMs, called $\mathbf{M}^4$, aiming to enhance LLMs' autonomous memory management capabilities through four key aspects: 1) Memory learning; 2) Memory construction; 3) Memory updating; 4) Memory retrieval. This essentially empowers LLMs with the ability to dynamically preserve and utilize memory information in long-term conversations.

**Category-based Memory Learning.** Inspired by the memory category that human memory mechanisms abstract historical information into different natural memory categories as memory clues (Mandler & Ritchey, 1977; Anderson, 2005), $\mathbf{M}^4$ designs a category-based memory extraction module to learn memory information from conversations. For each conversation session, unlike existing studies that coarsely summarize conversation into memory, $\mathbf{M}^4$ prompts LLMs to adaptively classify conversational information into distinct categories and retain information absent from their internal knowledge, which effectively prevents redundancy between the model's inherent knowledge and external memory components. This enables LLMs to autonomously identify what information and which types of knowledge need to be memorized.

**Chain-based Memory Construction.** Inspired by the method of loci (Yates, 1966), $\mathbf{M}^4$ converts abstract, disorganized, and easily forgotten information into a concrete, structured, and memorable spatial representation. Specifically, $\mathbf{M}^4$ represents each memory as a node and anchors the memory nodes in an orderly and spatially retrievable structure based on the chronological order of the conversation sessions. This enables LLMs to automatically organize different memories into ordered chains based on memory categories, called *memory chains*. Meanwhile, $\mathbf{M}^4$ can form memory graphs by linking shared memory nodes across memory chains. By drawing the evolutionary trajectory of the memories, this memory structure provides a foundation for dynamic memory retrieval.

**Self-monitoring-based Memory Updating.** For each memory chain, "*calibrate*" and "*compress*", two actions are used to perform self-monitoring in memory management, achieving memory consistency and efficiency while ensuring memory scalability. Specifically, $\mathbf{M}^4$ will perform two actions for each newly introduced memory information:

- *Calibrate*: To maintain memory accuracy and consistency, existing memory nodes are calibrated when a conflict arises with newly introduced information.
- *Compress*: To free storage space and reduce memory interference for incoming content, $\mathbf{M}^4$ compresses distant memory nodes upon the arrival of new information, making memory efficient.

**Self-reflection-based Memory Retrieval.** Inspired by heuristic search (Bonet & Geffner, 2001; Minsky, 2007), $\mathbf{M}^4$ performs dynamic and adaptive memory retrieval during query processing. It begins by selecting the most relevant memory node from the query-related category as the root node, then bidirectionally traverses associated memory chains along nodal connections to retrieve associative memory nodes, which are chronologically connected and activated to generate a response. At each retrieval step, $\mathbf{M}^4$ employs a self-reflective mechanism that enables LLMs to autonomously decide whether to adopt/skip the current memory node and continue to retrieve, or stop the retrieval process, based on the self-evaluation of the generated response.

To evaluate the effectiveness of $\mathbf{M}^4$ in memory management, we conduct extensive experiments on three benchmark datasets, including long-term conversation and long-term question-answering. Experimental results demonstrate that the proposed $\mathbf{M}^4$ can empower LLMs with self-memory capabilities, outperforming existing state-of-the-art memory management models in all dimensions of evaluation metrics and economizing storage space by more than 50%.

The contribution of this work can be summarized as follows:

- We propose meta-memory for memory management ($\mathbf{M}^4$), a novel framework that equips LLMs with the ability to adaptively and dynamically acquire and utilize memory in long-term conversations. This enables LLMs to self-judge for memory learning, self-organize for memory construction, self-monitor for memory update, and self-reflect for memory retrieval, significantly enhancing memory management for LLMs.
- We are the first to construct memory chains organized by category to model the evolutionary trajectories of memories. Building on this, we introduce a novel retrieval strategy that employs self-reflective chronological traversal, enabling dynamic and on-demand memory retrieval for response generation.
- Two actions, "*calibrate*" and "*compress*", are employed to enable self-monitoring within the memory management process, ensuring that memory remains comprehensive, unambiguous, and efficient.
- Experiments on automatic and human evaluations show that the proposed $\mathbf{M}^4$ significantly enhances the ability to take advantage of conversation memory and improves the quality of response generation for LLMs in long-term conversations.

## 2 RELATED WORK

Long-term conversation is an emerging task in open-domain conversation (Ritter et al., 2011; Li et al., 2017; Zhang et al., 2018; Dinan et al., 2018; Rashkin et al., 2019; Baumgartner et al., 2020; Thoppilan et al., 2022; Gu et al., 2023; Wen et al., 2023). It not only focuses on the long-term memory of a single session but also needs to consider time intervals between different sessions to enable long-term interaction. Xu et al. (2022a) first propose a multi-session chat dataset, where each session has a certain time interval, ranging from a few hours to a few days. Similarly, to reflect the

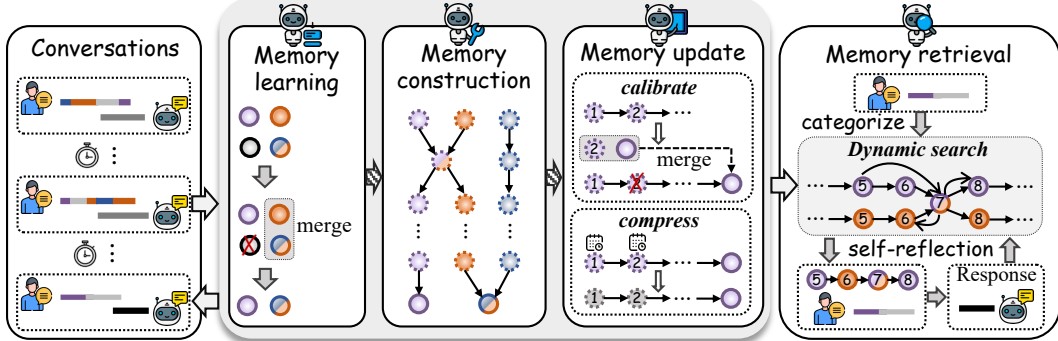

Figure 1: Illustration of our $\mathbf{M}^4$ framework. Colored circles represent memory nodes. Different colored circles represent different memory categories, and multiple colored circles represent nodes belonging to multiple categories. Circles with a solid line and with a dotted line represent memory nodes from the current conversation session and from previous conversation sessions, respectively. The numbers inside the circle represent the session number. ✗ represents deleting a node.

changes in information between sessions, Bae et al. (2022) propose a Korean multi-session conversation dataset. More recently, Jang et al. (2023) introduce Conversation Chronicles, which includes speaker relationships and a larger range of time intervals. However, these works primarily focus on curating datasets and selecting valuable conversational information to train generation models. A current trend is to build memory banks (Lu et al., 2023; Zhang et al., 2023; Zhong et al., 2024; Chen et al., 2025; Li et al., 2025; Ong et al., 2025; Wang et al., 2025) as plug-and-play modules for LLMs. Lu et al. (2023) propose self-composed memos for consistent conversation. Zhang et al. (2023) and Ong et al. (2025) pay more attention to the impact of the time interval on generation. Zhong et al. (2024) design memory forgetting and updating mechanisms to maintain long-term interactions. Recently, Chen et al. (2025) and Li et al. (2025) compress sessions into conversation summaries and user-specific facts. Wang et al. (2025) update memories iteratively by summarizing them, which can lead to the accumulation of errors and loss of nuanced information. **Unlike these simplistic update/forgetting mechanisms or iterative summarization studies that are prone to error accumulation, our $\mathbf{M}^4$ introduces an adaptive and dynamic framework for long-term memory management by integrating meta-memory into LLMs, providing a novel mechanism for on-demand memory storage and utilization.**

## 3 METHODOLOGY

This section provides a detailed description of the novel Meta-Memory for Memory Management ($\mathbf{M}^4$) framework. As shown in Figure 1, $\mathbf{M}^4$ mainly consists of four modules: 1) **Category-based Memory Learning**, which autonomously identifies what information and which types of knowledge should be retained from the conversation history. 2) **Chain-based Memory Construction**, which organizes memories into ordered chains according to their categories and constructs memory graphs based on shared memory nodes. 3) **Self-monitoring-based Memory Updating**, which performs self-monitoring for memory management by two actions: "*calibrate*" and "*compress*". 4) **Self-reflection-based Memory Retrieval**, which introduces self-reflection into memory retrieval to dynamically connect and integrate memory clues in response to specific queries on demand.

### 3.1 CATEGORY-BASED MEMORY LEARNING

For each conversation session, to determine which information in the historical conversation is missing from the specific LLM and needs to be memorized, $\mathbf{M}^4$ first extracts candidate categories $\mathcal{T} = \{\boldsymbol{t}_i\}_{i=1}^{n_o}$ from the conversational content. Where $\boldsymbol{t}_i \in \mathbb{R}^{d_t}$ is the embedding of the memory category. Further, to eliminate the ambiguity of memory categories, we perform cosine similarity on each pair of memory categories $(\boldsymbol{t}_i, \boldsymbol{t}_j)$ and merge categories with high similarity:

$$\eta = sim(\boldsymbol{t}_i, \boldsymbol{t}_j) = \frac{\boldsymbol{t}_i \cdot \boldsymbol{t}_j}{\|\boldsymbol{t}_i\|\|\boldsymbol{t}_j\|} \tag{1}$$

If $\eta >= \gamma$, categories $\boldsymbol{t}_i$ and $\boldsymbol{t}_j$ are merged as the same category $\boldsymbol{t}_i$, otherwise $\boldsymbol{t}_i$ and $\boldsymbol{t}_j$ are regarded as two independent categories. $\gamma$ is the similarity threshold. Then, for each memory category $\boldsymbol{t}_i$, $\mathbf{M}^4$ summarizes the conversational content to obtain a corresponding memory clue $\boldsymbol{c}^i \in \mathbb{R}^{d_c}$, which retains the key information of $\boldsymbol{t}_i$ in the session. Where $d_c$ is the dimension of clue embedding. Building on this, we can obtain a set of memory categories $\mathcal{T} = \{\boldsymbol{t}_i\}_{i=1}^n$ and corresponding memory clues $\{\boldsymbol{c}^i\}_{i=1}^n$, providing the fundamental disordered source for memory construction. Section E.1 shows examples of memory construction's prompting.

## 3.2 CHAIN-BASED MEMORY CONSTRUCTION

The essence of memory utilization is the effective extraction, integration, and application of stored information (Baddeley, 1983). However, when faced with a large amount of conversation history, it is difficult to obtain an optimal memory integration for the current conversation based on the extracted disordered memory clues, which essentially constitutes an NP-hard problem. To address this challenge, $\mathbf{M}^4$ introduces memory chains to store and manage memory clues, which is a method of reducing the computational consumption of disordered memory clue combinations by introducing ordered memory. This method originates from the "method of loci"(Anderson, 2005), which states that remembering an ordered sequence of items is important for memory management and utilization. At the same time, memory chains can effectively preserve the evolutionary trajectory of memory, enabling better memory activation for utilization.

For each memory category $\boldsymbol{t}^k$ and the corresponding memory clue $\boldsymbol{c}_{s_t}^k$ in the new session $s_t$, we first use Eq. 1 to calculate the similarity between $\boldsymbol{t}^k$ and each existing memory category $\boldsymbol{t}_i$ to obtain the similarity $\eta_i = sim(\boldsymbol{t}^k, \boldsymbol{t}_i)$. Therefore, the maximum similarity $\eta^k$ and the corresponding index $\mathcal{I}^k$ can be obtained based on the set of similarities:

$$\eta^k = \max\{\eta_i\}_{i=1}^{n^t}, \quad \mathcal{I}^k = \operatorname{argmax}\{\eta_i\}_{i=1}^{n^t} \tag{2}$$

Where $n^t$ represents the number of existing memory categories. Based on this, the new memory clue $\boldsymbol{c}_{s_t}^k$ serves as a node, which will be used to linked to the corresponding category-based memory chain $\mathcal{C}^{\mathcal{I}^k}$ or create a new memory chain for category $\boldsymbol{t}^k$:

$$\mathcal{C}^{\mathcal{I}^k} \leftarrow \begin{cases} f_c(\mathcal{C}^{\mathcal{I}^k}||\boldsymbol{c}_{s_t}^k) & \text{if } \eta^k >= \gamma, \\ \langle \boldsymbol{c}_{s_t}^k \rangle & \text{otherwise.} \end{cases} \tag{3}$$

Where $f_c(i||j)$ represents linking $j$ to the end of $i$. Here, nodes belonging to multiple categories will serve as shared memory to construct memory graphs, which will be used to activate memory clues for multiple categories simultaneously when responding to complex questions.

## 3.3 SELF-MONITORING-BASED MEMORY UPDATING

As the conversation progresses, memory information will continue to develop. Therefore, $\mathbf{M}^4$ introduces two actions, "*calibrate*" and "*compress*", to make memory consistent and efficient while ensuring scalability.

**Calibrate.** For each newly added memory node $\boldsymbol{c}_{s_t}^k$, $\mathbf{M}^4$ prompts LLMs to perform a consistency check across the memory chain $\mathcal{C}^k$, aiming to identify any factual conflicts, updates to previously stated information, or significant event evolutions. This action is defined as follows:

$$\boldsymbol{c}_u^k \leftarrow \text{LLM}_u(\mathcal{C}^k; \boldsymbol{c}_{s_t}^k), \quad \hat{\boldsymbol{c}}_{s_t}^k \leftarrow f_m(\boldsymbol{c}_{s_t}^k; \boldsymbol{c}_u^k), \quad \mathcal{C}^k \leftarrow f_u(\mathcal{C}^k; \boldsymbol{c}_u^k), \quad \mathcal{C}^k \leftarrow f_c(\mathcal{C}^k||\hat{\boldsymbol{c}}_{s_t}^k) \tag{4}$$

Where $\text{LLM}_u(\cdot; \cdot)$ represents consistency check. $\boldsymbol{c}_u^k$ represents inconstant memory, which can be one or more nodes, or it can be empty. $f_m(i; j)$ represents merging nodes $i$ and $j$ by LLMs to get

an unambiguous memory node. $f_u(i; j)$ represents removing memory node $j$ from memory chain $i$. This "calibrate" action ensures that each memory chain maintains internal consistency and reflects the most current state of information related to its specific category. Section E.3 and E.4 show examples of the prompting for the "calibrate" action.

**Compress.** Inspired by Ebbinghaus Forgetting Curve (Ebbinghaus, 2013), which states that human memory will decay to a stable level after more than a few weeks, we design a "compress" action to make room for new memory and alleviate the interference of distant information when using memory. Specifically, for a memory node $c_i$ that has not been used for more than a few weeks, we employ LLMLingua (Pan et al., 2024), a compressor to keep the important tokens for it and derive a compressed representation:

$$c_i^{com} = \text{LLMLingua}(c_i) \tag{5}$$

Note that this action is iterative, and the "few weeks" interval is provided by the dataset; otherwise, we will execute this action at one-week intervals. That is, multiple compressing actions will be performed on a memory clue that has not been used for a long time. However, as described in Ebbinghaus Forgetting Curve, human memory will decay rather than disappear completely. Therefore, in the iterative compression process, we introduce LLM to automatically determine whether the compressed memory node $c_i^{com}$ can recover the main tokens to understand the memory information. When LLMs are unable to recover the key meanings, the compression will be stopped, and the compressed memory node from the previous step is regarded as the final representation of this memory. Section E.5 shows examples of the prompting for the "compress" action.

### 3.4 Self-reflection-based Memory Retrieval

Unlike existing work that directly uses query-related memory information when generating responses, $\mathbf{M}^4$ introduces a dynamic memory retrieval algorithm to retrieve and integrate contributory memories for response generation. For each query $q$, we first perform prompting to obtain the set of memory categories $\mathcal{K}^q$, and then use Eq. 1 to obtain a list of memory categories $\mathcal{T}^q$ that exist in memory chains. Afterwards, when the LLM lacks confidence in responding to the current query $q$, $\mathbf{M}^4$ conducts the following steps:

**Step 1:** For each memory category $t_i^q$ in $\mathcal{T}^q$, we use Eq. 1 to obtain the most relevant memory clue $c_k^q$ from memory chain $\mathcal{C}^q$ of the memory category $t_i^q$ and integrate it to the retrieved memory chain $\mathcal{M}: \mathcal{M} \leftarrow <c_k^q>$.

**Step 2:** Starting from $c_k^q$, we perform a gradual bidirectional search through neighboring memory nodes, integrating a node into $\mathcal{M}$ if the self-evaluation of LLM determines that it would improve the response quality:

$$\mathcal{M} \leftarrow \begin{cases} f_c(\mathcal{M}||c_{k+1}^q) & \text{if } \text{LLM}_{re}(c_{k+1}^q), \\ f_c(c_{k-1}^q||\mathcal{M}) & \text{if } \text{LLM}_{re}(c_{k-1}^q). \end{cases} \tag{6}$$

Where $\text{LLM}_{re}(i)$ represents that the integration of memory node $i$ into $\mathcal{M}$ can improve the response quality according to the self-evaluation of LLM. Note that memory remains in its original order during integration, which allows for perception of the evolution of memory categories and characterization of memory traces. The retrieved memory $\mathcal{M}$ is fed into LLMs for response generation when $\mathcal{M}$ is updated each time:

$$r = \text{LLM}(q, \mathcal{M}) \tag{7}$$

The retrieval terminates when LLMs deem that the current $\mathcal{M}$ is confident to respond to query $q$ without laboriously exploring the entire memory space. The procedure of Self-reflection-based Memory Retrieval is depicted in Algorithm 1. In addition, when the memory chains of all relevant categories have been traversed but cannot answer the current query, LLM will automatically locate the shared node and continue to retrieve relevant memory clues according to the above steps. Section E.7 shows examples of the prompting for response generation, which prompts LLMs to pay more attention to the order and evolution of the memorized explicitly.

Table 1: Automatic evaluation of response quality (average of sessions). "**Bold Font**" means the highest results. "Context" denotes feeding history information directly into the long context of LLMs. *B-4 = BLEU-4, R-L = ROUGE-L, and Bert = BertScore.

| Backbone | Methods | CC | | | | MSC | | | |
|---|---|---|---|---|---|---|---|---|---|
| | | B-4 | R-L | Mauve | Bert | B-4 | R-L | Mauve | Bert |
| Qwen2.5 | Context | 1.31 | 14.52 | 54.72 | 45.71 | 1.05 | 13.83 | 54.31 | 48.23 |
| | MemoChat (Lu et al., 2023) | 0.74 | 11.13 | 47.88 | 44.23 | 0.66 | 11.21 | 54.02 | 46.50 |
| | MemoryBank (Zhong et al., 2024) | 1.19 | 14.00 | 53.25 | 45.85 | 0.96 | 13.25 | 55.72 | 48.08 |
| | COMEDY (Chen et al., 2025) | 0.69 | 11.38 | 47.71 | 44.92 | 0.64 | 11.21 | 55.25 | 47.07 |
| | Rsum (Wang et al., 2025) | 1.05 | 13.65 | 48.91 | 45.88 | 0.83 | 12.84 | 56.61 | 47.86 |
| | THEANINE (Ong et al., 2025) | 1.04 | 13.57 | 59.99 | 44.47 | 1.02 | 13.32 | 41.57 | 47.20 |
| | $M^4$ (Ours) | **1.90** | **16.73** | **69.54** | **46.81** | **1.26** | **14.15** | **64.88** | **48.29** |
| Llama3 | Context | 1.08 | 13.60 | 57.75 | 45.25 | 0.89 | 13.63 | 59.34 | 47.26 |
| | MemoChat (Lu et al., 2023) | 0.81 | 11.43 | 30.19 | 43.42 | 0.67 | 11.95 | 48.45 | 46.08 |
| | MemoryBank (Zhong et al., 2024) | 0.81 | 12.27 | 53.94 | 45.45 | 0.70 | 12.20 | 59.86 | 46.91 |
| | COMEDY (Chen et al., 2025) | 0.54 | 9.45 | 48.12 | 44.03 | 0.53 | 10.32 | 55.86 | 45.94 |
| | Rsum (Wang et al., 2025) | 1.10 | 13.86 | 47.89 | 45.21 | 0.70 | 12.52 | 59.51 | 47.34 |
| | THEANINE (Ong et al., 2025) | 0.67 | 11.83 | 47.19 | 44.35 | 0.72 | 12.59 | 53.81 | 47.10 |
| | $M^4$ (Ours) | **2.00** | **17.79** | **66.63** | **45.60** | **1.09** | **14.81** | **64.02** | **47.52** |
| ChatGPT | Context | 2.36 | 17.85 | 67.82 | 48.27 | 1.20 | 14.83 | 59.24 | 47.39 |
| | MemoChat (Lu et al., 2023) | 1.71 | 15.22 | 50.98 | 46.13 | 1.04 | 13.52 | 58.32 | 46.57 |
| | MemoryBank (Zhong et al., 2024) | 0.78 | 10.79 | 46.90 | 42.53 | 0.59 | 10.15 | 54.77 | 43.75 |
| | COMEDY (Chen et al., 2025) | 0.92 | 12.55 | 41.33 | 45.77 | 0.76 | 12.43 | 54.28 | 46.99 |
| | Rsum (Wang et al., 2025) | 1.16 | 13.86 | 50.58 | 45.70 | 0.78 | 11.85 | 57.53 | 45.69 |
| | THEANINE (Ong et al., 2025) | 1.01 | 14.40 | 57.55 | 45.41 | 1.12 | 13.99 | 45.18 | 47.75 |
| | $M^4$ (Ours) | **3.00** | **19.49** | **78.26** | **48.88** | **1.36** | **15.36** | **65.13** | **48.37** |
| GPT-4o | Context | 1.79 | 17.41 | 55.11 | 47.79 | 1.21 | 15.12 | 54.51 | 49.17 |
| | MemoChat (Lu et al., 2023) | 1.71 | 15.22 | 50.98 | 46.13 | 0.83 | 12.63 | 53.41 | 47.93 |
| | MemoryBank (Zhong et al., 2024) | 1.08 | 15.14 | 45.95 | 47.27 | 1.03 | 13.74 | 45.30 | 48.39 |
| | COMEDY (Chen et al., 2025) | 0.67 | 11.30 | 39.51 | 46.18 | 0.60 | 11.07 | 48.86 | 47.19 |
| | Rsum (Wang et al., 2025) | 1.01 | 14.57 | 46.51 | 47.12 | 0.97 | 13.99 | 51.99 | 48.43 |
| | THEANINE (Ong et al., 2025) | 1.25 | 14.42 | 56.13 | 45.19 | 0.94 | 13.57 | 54.14 | 47.52 |
| | $M^4$ (Ours) | **2.03** | **18.07** | **66.59** | **47.84** | **1.29** | **15.39** | **58.22** | **49.08** |

# 4 EXPERIMENTS

In this section, we first evaluate the performance of the proposed $M^4$ compared with existing state-of-the-art (SOTA) baselines in long-term conversation. Then, we present a detailed analysis of our $M^4$ to demonstrate its effectiveness in memory management for LLMs.

## 4.1 EXPERIMENTAL SETTINGS

**Datasets.** We conduct extensive experiments on two long-term conversation datasets: **MSC** (Xu et al., 2022a) and **CC** (Jang et al., 2023), each comprising 5 sessions with approximately 50 conversational turns per sample. Moreover, we also evaluate the robustness for long-term memory capacity of $M^4$ on a long-term question-answering at **LONGMEMEVAL** (Wu et al., 2025) dataset. Appendix A for more details about datasets.

**Models and Baselines.** We evaluate on four mainstream LLMs: 1) **Qwen2.5-7B** (Yang et al., 2024), the Qwen2.5-7B-Instruct version. 2) **Llama3-8B** (Touvron et al., 2023), the Meta-Llama-3-8B-Instruct version. 3) **ChatGPT** (OpenAI, 2023), the GPT-3.5-Turbo-0125 version. 4) **GPT-4o** (OpenAI, 2024a), the GPT-4o-2024-08-06 version. For response generation and question answering, we compare our method with five strong baselines: **MemoChat** (Lu et al., 2023), **MemoryBank**

(Zhong et al., 2024), **Rsum** (Wang et al., 2025), **COMEDY** (Chen et al., 2025), **THEANINE** (Ong et al., 2025), and **LMEBOT** (Wu et al., 2025). For the embedding model, we use text-embedding-3-large (OpenAI, 2024b). More details of baselines are shown in Appendix B.

In addition, we design various variants to present an ablation study of our $\mathbf{M}^4$: 1) To study the effectiveness of the proposed chain-based memory construction, we provide "w/o Chain" to construct memories in random order. 2) To evaluate the effectiveness of the Self-monitoring-based Memory Updating, we provide "w/o Calibrate" and "w/o Compress" to analyze the importance of these two actions. 3) We provide "w/o Self-reflection" as directly using all query-related memory in response generation without self-reflection during memory retrieval.

**Evaluation Metrics.** We comprehensively evaluate our $\mathbf{M}^4$ on three types of metrics. 1) **Automatic Metrics.** Following Ong et al. (2025), we use BLEU-4 (Papineni et al., 2002), ROUGE-L (LIN, 2004), Mauve (Pillutla et al., 2021), and BertScore (Zhang et al., 2019) to automatically evaluate response generation. Following Wu et al. (2025), accuracy is used for comparison with LMEBOT on LONGMEMEVAL. 2) **G-Eval Metrics.** Following Xu et al. (2022b) and Jang et al. (2023), we use GPT-4o to evaluate dialogue generation on five dimensions: *Engagingness*, *Humanness*, *Coherence*, *Consistency*, and *Memorability*. Appendix D for details about these metrics. 3) **Human Metrics.** Following Ong et al. (2025), humans evaluate the winning performance of different methods on response generation and memory retrieval.

## 4.2 EXPERIMENTAL RESULTS

**Main results.** The experimental results presented in Table 1 compellingly demonstrate the superior performance of our proposed $\mathbf{M}^4$ framework in long-term conversation. Our $\mathbf{M}^4$ framework consistently outperform all baseline models on both long-term conversation datasets across all evaluation metrics. Further, $\mathbf{M}^4$ outperforms all the memory management baselines, demonstrating that endowing LLMs with autonomous memory management ability by exploring meta-memory is more effective in long-term information learning compared to laborious mining of memory information from historical conversations.

**Ablation Study.** To further analyze the impact of each module of our $\mathbf{M}^4$ on performance, we conduct an ablation study on various variants of $\mathbf{M}^4$ with the ChatGPT backbone and report the results in Table 2. The removal of the memory chain ("w/o Chain") significantly decreases the performance on all metrics. This indicates that the inherent order within memory chains enables the LLM to better understand temporal dependencies, narrative flow, and the evolution of memory, leading to more contextually appropriate responses compared to a disordered set of memories. Further, ablation of

Table 2: Ablation study of $\mathbf{M}^4$ on ChatGPT.

| Datasets | Methods | B-4 | R-L | Mauve | Bert |
|---|---|---|---|---|---|
| CC | $\mathbf{M}^4$ (Ours) | **3.00** | **19.49** | **78.26** | **48.88** |
| | w/o Chain | 2.40 | 17.67 | 72.57 | 48.30 |
| | w/o Calibrate | 2.89 | 19.28 | 76.37 | 48.77 |
| | w/o Compress | 2.90 | 19.06 | 74.38 | 48.79 |
| | w/o Self-reflection | 2.90 | 19.18 | 76.79 | 49.26 |
| MSC | $\mathbf{M}^4$ (Ours) | **1.36** | **15.36** | **65.13** | **48.37** |
| | w/o Chain | 1.24 | 14.88 | 62.41 | 48.07 |
| | w/o Calibrate | 1.27 | 15.00 | 63.35 | 48.11 |
| | w/o Compress | 1.26 | 14.97 | 64.21 | 48.10 |
| | w/o Self-reflection | 1.31 | 15.21 | 64.73 | 48.14 |

either the "calibrate" or "compress" action results in a considerable performance degradation. This denotes that, compared to static memory storage, updating stored memories appropriately can improve memory management and thus enhance text generation for long-term conversations. The inferior performance of the "w/o Self-reflection" variant highlights the importance of self-reflection in dynamically retrieving relevant memories, which facilitates memory learning and elicits higher-quality responses compared to the direct use of stored memories.

## 4.3 ANALYSIS OF OUR $\mathbf{M}^4$

$\mathbf{M}^4$ **improves both LLMs' memory management and response quality.** To further evaluate the quality of response generation beyond automatic metrics, we perform a comprehensive G-Eval (Liu et al., 2023) on five key dimensions. As shown in Figure 2 (a) and (b), compellingly demonstrate the superior performance of our proposed method compared to six baselines. Notably, our method exhibits substantial advantages in *Memorability*, suggesting its effectiveness in recalling and appropriately utilizing long-term historical information. It also excels in *Engagingness*, *Coherence*, and

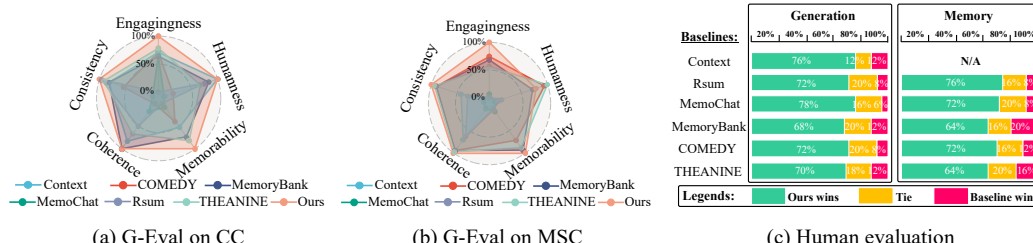

(a) G-Eval on CC  (b) G-Eval on MSC  (c) Human evaluation

Figure 2: Evaluation of response quality by G-Eval metrics (a and b) and human evaluation (c).

*Consistency*, crucial for maintaining logical and contextually sound conversations. In addition, we additionally implement a human evaluation. As shown in Figure 2 (c), the human evaluation results overwhelmingly favor our $\mathbf{M}^4$. In terms of "Generation" quality, $\mathbf{M}^4$ are preferred by human evaluators in a significant majority of cases. The winning rate exceeds 68% across all the baselines. For the evaluation of "Memory" utilization, $\mathbf{M}^4$ demonstrates significant advantages compared with the baseline models. Human evaluation demonstrates that our $\mathbf{M}^4$ not only excels in automatic metrics but also produces responses that are qualitatively superior in terms of both generation quality and the effective use of long-term conversational memory.

**Memory categories can enhance the ability of LLM to retrieve relevant memory information.** To further explore the benefit of the proposed $\mathbf{M}^4$ in enhancing the capability of memory utilization, we manually select three control groups for the experiment, each group consisting of 10 retrieved memory embedding combinations based on queries. The topics mentioned in the queries of the two groups are similar, while the remaining group is unrelated to these two groups. The t-SNE visualization in Figure 3 reveals a clear distinction: retrieved memories for content-related queries form coherent clusters, while those for unrelated queries are widely dispersed. This indicates that memories retrieved by content-related queries are highly correlated, while those from unrelated queries are uncorrelated. That is, $\mathbf{M}^4$ can dynamically retrieve task-relevant memories for different queries, leading to more accurate and contextually appropriate responses.

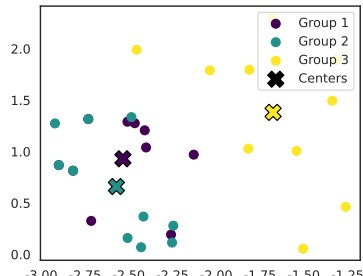

Figure 3: t-SNE visualization of memories retrieved based on different queries. Queries in groups 1 and 2 contain similar topics, while group 3 involves different topics.

**The "compress" action can substantially reduce token consumption for memory storage.** Figure 4 illustrates the memory usage (tokens per conversation) of our $\mathbf{M}^4$ with and without the "compress" action on both long-term conversation datasets. The results clearly show a substantial reduction in token consumption when compressing is applied, decreasing the memory space by more than 50% on both datasets. This significant decrease in token requirements highlights the effectiveness of our Compressing action in creating concise yet informative memory representations, which is crucial for managing long conversational histories efficiently, especially given the token limits and computational costs associated with LLMs.

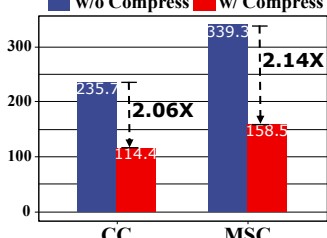

Figure 4: Memory usage comparison of ChatGPT.

**Parameter Sensitivity of $\gamma$.** We further analyze the sensitivity of our $\mathbf{M}^4$ framework to the similarity threshold $\gamma$, a crucial parameter governing the relevance assessment for memory linking, with results presented in Figure 5. The performance only fluctuates slightly for different values, indicating that $\mathbf{M}^4$ can fit different similarity thresholds when computing the relevant memory information. Further, the results suggest that a moderately selective similarity threshold strikes an effective balance without being overly lenient (introducing noise) or overly strict (missing useful context).

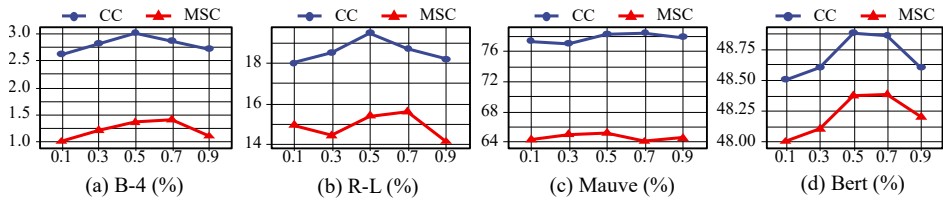

Figure 5: Experimental results of parameter sensitivity analysis about similarity threshold $\gamma$.

**Analysis of task robustness.** To further analyze the robustness of $\mathbf{M}^4$ in other long-term information learning tasks. We experiment on a long-term question-answering dataset and report the results in Table 3. We can see that $\mathbf{M}^4$ consistently outperforms both the Context-only baseline and LMEBOT across all three challenging tasks. This demonstrates the efficacy of $\mathbf{M}^4$ in utilizing memory dispersed across multiple conversational turns. For the knowledge update task, this strong result underscores the effectiveness of our Memory Evolution action in maintaining an accurate and up-to-date knowledge base. Moreover, the result of temporal reasoning indicates that the inherent sequential or-

Table 3: Question answering accuracy (%) on LONGMEMEVAL compared to baselines. MR = Multi-Session Reasoning, KU = Knowledge Update, and TR = Temporal Reasoning.

| Backbone | Methods | MR | KU | TR |
|---|---|---|---|---|
| | Context | 47.38 | 50.00 | 40.87 |
| Qwen2.5 | LMEBOT (Wu et al., 2025) | 51.88 | 56.41 | 45.11 |
| | $\mathbf{M}^4$ (Ours) | **53.89** | **64.10** | **51.88** |
| | Context | 49.62 | 53.85 | 47.37 |
| Llama3 | LMEBOT (Wu et al., 2025) | 53.38 | 64.95 | 54.89 |
| | $\mathbf{M}^4$ (Ours) | **55.41** | **67.95** | **56.39** |
| | Context | 31.58 | 35.90 | 33.08 |
| ChatGPT | LMEBOT (Wu et al., 2025) | 54.86 | 53.85 | 40.60 |
| | $\mathbf{M}^4$ (Ours) | **58.65** | **57.69** | **43.60** |

ganization of our memory chains aids in understanding and reasoning about the temporal relationships between events and information presented across different sessions.

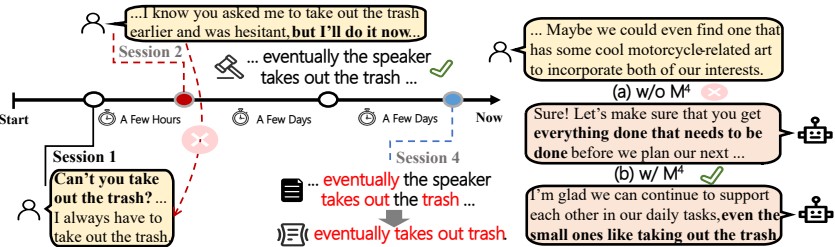

Figure 6: Case study on ChatGPT compared to w/ and w/o $\mathbf{M}^4$.

**Case Study.** Figure 6 depicts a multi-session scenario where a user complaint in Session 1 is resolved in Session 2 when the speaker agrees to do so. Our $\mathbf{M}^4$ framework, through its Memory Evolution action, updates the relevant memory chain to reflect this resolution (symbolized by "*...eventually the speaker takes out the trash...*"). Subsequently, the Compressing action distills this evolved memory into its core essence ("*eventually takes out trash*") for efficient long-term storage. Moreover, $\mathbf{M}^4$ leverages the accurately evolved and efficient memory, generates a response ("*... even the small ones like taking out the trash*") that not only addresses the current interaction but also subtly acknowledges the past resolved issue.

## 5 CONCLUSIONS

In this paper, we propose $\mathbf{M}^4$, a novel framework for LLMs' long-term memory learning. By leveraging meta-memory into memory management, $\mathbf{M}^4$ enables LLMs to dynamically manage and utilize memory during response generation. Through comprehensive experiments on multiple benchmarks, we demonstrate that $\mathbf{M}^4$ consistently outperforms existing baselines in long-term conversation tasks. Further analysis shows that through the introduction of self-monitoring and self-reflection mechanisms, our $\mathbf{M}^4$ achieves superior memory utilization and generates higher-quality responses.

**Ethical Statement** This research strictly adheres to data usage regulations; all experiments are based on public datasets, with a commitment not to process any private information. While the current work does not delve into ethical topics like transparency and inclusivity, we acknowledge their value and believe future advancements can be integrated into our memory learning framework. We encourage the academic community to work together to enhance the understanding and implementation of responsible AI.

**Reproducibility Statement** To facilitate the reproduction of our results, Section 4 and the Appendix provide a thorough description of our experimental setup, evaluation metrics, and implementation specifics. The source code and scripts will be made publicly available upon this paper's acceptance. We have also listed all required external libraries and dependencies. To demonstrate the broad applicability of our approach, we have validated it on both open-source and commercial models.

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

## A   APPENDIX

## A   MORE DETAILS OF DATASETS

| Datasets | # Sessions (train) | # of Sessions | Time Interval | | | | | Relation |
|---|---|---|---|---|---|---|---|---|
| | | | Hours | Days | Weeks | Months | Years | |
| MSC (Xu et al., 2022a) | 4 | 16K | ✓ | ✓ | ✗ | ✗ | ✗ | ✗ |
| CC (Jang et al., 2023) | 5 | 1M | ✓ | ✓ | ✓ | ✓ | ✓ | ✓ |

Table 4: Comparison between MSC and CC. The latter considers longer time intervals and speaker relationships.

We evaluate our method on two long-term multi-session conversation datasets: **Conversation Chronicles** (CC) Jang et al. (2023) and **Multi-Session Chat** (MSC) Xu et al. (2022a). Following Ong et al. (2025)'s settings, we randomly select 50 episodes from the test set of each dataset, a total of 250 sessions for generation experiments in this paper. The statistics of each data set are shown in Table 4. For additionally **LONGMEMEVAL** dataset, which focuses on question answering, we select full LONGMEMEVAL$_S$ (50 sessions per question, total 500 questions) for our experiments to evaluate three subtasks: Multi-Session Reasoning, Knowledge Update, and Temporal Reasoning.

## B   COMPARED BASELINES

There are five strong summary-based baselines in the paper for comparison with our method:

- **MemoChat** (Lu et al., 2023): This work summarizes different topics separately and stores them in memory by constructing structured memos.

- **MemoryBank** (Zhong et al., 2024): This work creates a memory bank based on the Eisen-haus forgetting curve to manage the memory of user portraits and summaries.

- **COMEDY** (Chen et al., 2025): This work uses user profiles, relationship descriptions, and events from past conversations as compressed summaries to prompt LLMs (i.e, ChatGPT).

- **Rsum** (Wang et al., 2025): This work uses LLM itself to iteratively summarize past conversations as memory to store. Specifically, after each summary, the old memory and the current context are summarized into a new memory.

- **THEANINE** (Ong et al., 2025): This work manages memories by linking them into time-lines based on temporal and cause-effect relationships, instead of deleting old ones.

For a fair comparison with our approach, we select the same environment named LMDeploy (Contributors, 2023) for inference on Qwen2.5-7B and Llama3-8B. For ChatGPT, we call OpenAI's API service for inference. We set temperature to 0.80 and $\gamma$ to 0.5 for generation.

## C   THE PROCEDURE OF SELF-REFLECTION-BASED MEMORY RETRIEVAL

We have included pseudocode (Algorithm 1) to ensure that readers can readily grasp the intricate details and flow of our algorithm.

## D   G-EVAL METRICS

With the development of open-domain conversation based on LLM, traditional overlap metrics such as BLEU (Papineni et al., 2002), ROUGE (LIN, 2004), etc. face great challenges. The reason is that a wide range of response generation can be considered as appropriate responses (Liu et al., 2016). To this end, we refer to G-Eval (Liu et al., 2023) and use GPT-4o to evaluate episodes. In our paper, we follow the metrics set in Xu et al. (2022b) and Jang et al. (2023):

---

**Algorithm 1** Self-reflection-based Memory Retrieval

---

**Input:** Query $q$, query-related category set $\mathcal{T}^q = \{\boldsymbol{t}_i^q\}_{i=1}^{n^q}$ and corresponding memory chain set $\{\mathcal{C}_i^q\}_{i=1}^{n^q}$, self-evaluation function of the response $\text{LLM}_{\text{confident}}(\cdot)$, self-evaluation function of the newly retrieved node $\text{LLM}_{re}(\cdot)$, traversal pointers $p_f$ and $p_b$ for each memory chain

**Output:** Retrieved memory $\mathcal{M}$

1: $\mathcal{M} \leftarrow \emptyset$
2: **for all** $\boldsymbol{t}_i^q \in \mathcal{T}^q$ **do**
3: $\quad \mathcal{A} \leftarrow \text{LLM}_{\text{confident}}(q, \mathcal{M})$ # Determine whether LLM is confident in the current response
4: $\quad$ **if** $\mathcal{A} ==$ True **then**
5: $\qquad$ **Break**
6: $\quad k \leftarrow \text{argmax}\big\{sim(\boldsymbol{c}_j^q, q)\big\}_{j=1}^{|\mathcal{C}_i^q|}$ # Obtain the index of the most relevant memory clue by computing the similarity between each $\boldsymbol{c}_j^q$ and $q$
7: $\quad \mathcal{M} \leftarrow f_c(\mathcal{M}||\boldsymbol{c}_k^q)$ # Integrate $\boldsymbol{c}_k^q$ into $\mathcal{M}$
8: $\quad p_f \leftarrow k+1, \quad p_b \leftarrow k-1$ # Initialize the index of traversal pointers
9: $\quad \mathcal{A} \leftarrow \text{LLM}_{\text{confident}}(q, \mathcal{M})$
10: $\quad$ **while** $\mathcal{A} ==$ False **do**
11: $\qquad$ **if** $p_f < |\mathcal{C}_i^q| - 1$ **then**
12: $\qquad\quad \mathcal{B}_f \leftarrow \text{LLM}_{re}(\boldsymbol{c}_{p_f}^q)$ # Determine whether the response quality is improved by integrating memory node $\boldsymbol{c}_{p_f}^q$ into $\mathcal{M}$
13: $\qquad$ **else**
14: $\qquad\quad \mathcal{B}_f \leftarrow$ False # All the forward nodes in memory chain $\mathcal{C}_i^q$ have been traversed
15: $\qquad$ **if** $p_b > 0$ **then**
16: $\qquad\quad \mathcal{B}_b \leftarrow \text{LLM}_{re}(\boldsymbol{c}_{p_b}^q)$ # Determine whether the response quality is improved by integrating memory node $\boldsymbol{c}_{p_b}^q$ into $\mathcal{M}$
17: $\qquad$ **else**
18: $\qquad\quad \mathcal{B}_b \leftarrow$ False # All the backward nodes in memory chain $\mathcal{C}_i^q$ have been traversed
19: $\qquad$ **if** $\mathcal{B}_f ==$ True **then**
20: $\qquad\quad \mathcal{M} \leftarrow f_c(\mathcal{M}||\boldsymbol{c}_{p_f}^q)$
21: $\qquad\quad \mathcal{A} \leftarrow \text{LLM}_{\text{confident}}(q, \mathcal{M})$
22: $\qquad$ **if** $\mathcal{B}_f ==$ True **then**
23: $\qquad\quad \mathcal{M} \leftarrow f_c(\boldsymbol{c}_{p_b}^q||\mathcal{M})$
24: $\qquad\quad \mathcal{A} \leftarrow \text{LLM}_{\text{confident}}(q, \mathcal{M})$
25: $\qquad$ **if** $p_f == |\mathcal{C}_i^q| - 1$ and $p_b == 0$ **then**
26: $\qquad\quad$ **Break** # All the nodes in memory chain $\mathcal{C}_i^q$ have been traversed
27: $\qquad$ **if** $p_f < |\mathcal{C}_i^q| - 1$ **then**
28: $\qquad\quad p_f \leftarrow p_f + 1$
29: $\qquad$ **if** $p_b > 0$ **then**
30: $\qquad\quad p_b \leftarrow p_b - 1$
31: **return** $\mathcal{M}$

---

- **Engagingness**: The assistant can have rich interactions with users that go beyond simple conversations. For example, the assistant can generate interesting and immersive responses based on the current context.

- **Humanness**: The assistant can communicate with users like a real human would, displaying emotional understanding like empathy and human thought processes.

- **Coherence**: The assistant can generate responses that match current and historical contexts based on the context.

- **Consistency**: The assistant need to maintain consistent responses with their persona in long-term conversations.

- **Memorability**: The assistant can correctly recall more what happened in past sessions.

Each metric is scored on a scale of 1-5, with 1 being the worst and 5 being the best. Normalisation is taken in G-Eval experimental results to maintain a better visualisation.

---

**Prompt for Category-based Memory**

"""
You are an AI assistant with a Category-based Memory module. Your task is to extract and categorize new information from conversations to remember for future interactions. This process is designed to prevent redundancy by focusing only on information absent from your general knowledge.

Here is a conversation:
{Conversation}

Please analyze the conversation and identify distinct pieces of information that are important to memorize.

Focus exclusively on details specific to the speakers or the situation (e.g., personal preferences, plans, specific events, relationships) and avoid summarizing common knowledge you already possess. This enables you to autonomously identify what needs to be memorized.

For each identified piece of memory, classify it into a distinct and natural memory category and provide a concise summary.

Return a JSON array of objects, where each object represents a memory item.

Example JSON Response:
"""
{"Category": "{YOUR_CATEGORY}", "Summary": "{YOUR_SUMMARY}"},
{"Category": "{YOUR_CATEGORY}", "Summary": "{YOUR_SUMMARY}"}

Now, for the given conversation, provide the JSON response without any reasoning:
"""

Figure 7: Prompt for Category-based Memory.

---

**Prompt for Category Retrieval**

"""
Given the current utterance/query:
{Utterance}
Please analyze the utterance and determine which categories it is relevant to.
Return a JSON array of objects, where each object represents a category that is relevant to the utterance.

Example JSON Response:
"""
{"Category": "{YOUR_CATEGORY}"},
{"Category": "{YOUR_CATEGORY}"}

Based on the utterance, please provide the JSON response with the relevant categories without any reasoning:
"""

Figure 8: Prompt for Category Retrieval.

---

**Prompt for Conflict Detection**

"""
The following are summaries given in conversational order:
Previous_summary:
{Previous_summary} New_summary:
{New_summary}
Please understand their differences carefully and judge whether there is a conflict or update between them, such as a change in the speaker's identity, the development of events, and a change in attitude (interest, etc.).
If you think there is a conflict between them, then please output 1, otherwise output 0 without any reasoning:
"""

---

Figure 9: Prompt for Conflict Detection.

---

**Prompt for Memory Updating**

"""
The following are some summaries given in conversational order:
Previous_summary:
{Previous_summary}
New_summary:
{New_summary}
Please understand their differences carefully and focus on the changes or updates between contents, such as the change of the speaker's identity, the development of events, and the change of attitude (interest, etc.), etc., and merge the two into a new summary without any reasoning:
"""

---

Figure 10: Prompt for Memory Updating.

---

**Prompt for Compression Detection**

"""
The following is an original text and its compressed text:
Original_text:
{Original_text}
Compressed_text:
{Compressed_text}
Please be careful to understand the differences and make an honest judgement as to whether most of the key information in the original text can be restored from the compressed text.
If you think you can decompress or restore it, then please output 1, otherwise output 0 without any reasoning:
"""

---

Figure 11: Prompt for Compression Detection.

## E    GENERATION PROMPTS

### E.1    CATEGORY-BASED MEMORY PROMPT

As shown in Figure 7, we prompt LLMs to complete the category-based memory of each historical session.

---

Prompt for Memory Judgment

"""
The following are user's query/utterance and candidate memories:
Query/Utterance:
{Query/Utterance}
Memories:
{Memories}
Please be very honest in your judgement as to whether you can generate the best response based on the known memories.
If you can please output 1 and conversely if you need more memories then output 0 without any reasoning:
"""

Figure 12: Prompt for Memory Judgment.

---

Prompt for Response Generation

"""
The following are user's query/utterance and related memories:
Query/Utterance:
{Query/Utterance}
Memories:
{Memories}
Generate the most plausible answer based memories without any reasoning. Each line in the memory represents a timeline chain of topics. Please pay attention to the sequence, the evolution of these memories.
Do not put too much information in the next response.
"""

Figure 13: Prompt for Response Generation.

### E.2 CATEGORY RETRIEVAL PROMPT

As shown in Figure 8, we prompt LLMs to select possible categories that are more relevant to the current utterance.

### E.3 CONFLICT DETECTION PROMPT

As shown in Figure 9, we prompt LLMs to check for conflicts in the memory before the undergoing session. If conflicts are found between memorizes of the same topic in different sessions, the memory needs to be updated.

### E.4 MEMORY UPDATTING PROMPT

As shown in Figure 10, we prompt LLMs to merge previous memories with present ones. The point is to fuse the contents as a new conflict-free memory.

### E.5 COMPRESSION DETECTION PROMPT

As shown in Figure 11, we prompt LLMs to determine whether most of the key information of the original text can be restored based on the currently compressed text.

---

**Prompt for G-Eval**

```
"""
This a conversation:
{Conversation}
You are an impartial evaluator. Please evaluate this conversation based on the following five
metrics:
1. Engagingness: Two speakers should interact to create responses that are not only interest-
ing but also well-immersed in the given context of the conversation.
2. Humanness: Two speakers should have a conversation that demonstrates emotional un-
derstanding (e.g., empathy) and the use of natural language and thought processes that are
typical of human beings.
3. Memorability: If two Speakers recall past events correctly by retaining information from
previous sessions.
4. Coherence: Whether the whole conversation is relevant and consistent with the context.
5. Consistency: Whether responses are relevant and consistent with previous persona.
The score for each metric is 1-5, with 1 being the lowest score and 5 being the highest score.
Write down your score for each metric without any explanation, e.g. Engagingness: {YOUR
SCORE}, Humanness: {YOUR SCORE}, Memorability: {YOUR SCORE}, Coherence:
{YOUR SCORE}, Consistency: {YOUR SCORE}"
"""
```

Figure 14: Prompt for G-Eval.

### E.6 Memory Judgment Prompt

As shown in Figure 12, we prompt LLMs to determine whether current memories can provide suf-
ficient information to respond based on the current query/utterance.

### E.7 Response Generation Prompt

As shown in Figure 13, we prompt LLMs to pay more attention to the order and evolution of the
memorizes explicitly.

## F Evaluation Prompt

As shown in Figure 14, we prompt LLMs to evaluate all generated responses according to the defined
metrics.

## G Limitations

Our work introduces $\mathbf{M}^4$, a novel paradigm for dynamic memory management in long-term conver-
sations, establishing a foundational framework for more autonomous AI systems. While our exper-
iments demonstrate the significant potential of this approach, we also recognize several promising
avenues for future exploration that can further enhance its capabilities and robustness.

The effectiveness of the $\mathbf{M}^4$ framework is closely intertwined with the underlying capabilities of the
base LLM, as it relies on the model's reasoning for critical sub-tasks such as memory categoriza-
tion, conflict detection, and self-reflective retrieval. Future research could explore the development
of more specialized and lightweight mechanisms for these components. This could not only improve
computational efficiency but also enhance the overall system's resilience by reducing the potential
for cascading errors originating from a single judgment by the base model. Furthermore, the cur-
rent implementation employs a set of effective heuristics for memory retrieval and organization. A
valuable next step would be to investigate adaptive methods that allow the model to dynamically
adjust these strategies based on the specific conversational context or task, moving towards a more
sophisticated and self-optimizing memory management system.

## H  THE USE OF LARGE LANGUAGE MODELS (LLMs)

During the preparation of this manuscript, we utilized LLMs to assist with proofreading and language polishing. The authors have reviewed and edited all suggested changes to ensure the scientific accuracy and clarity of the content, and take full responsibility for the final manuscript.

