# OpenReview forum: "Meta-Memory for Large Language Models"
_ICLR.cc/2026/Conference — ICLR 2026 Conference Withdrawn Submission_

### Official Review · Reviewer_9EQk · 2025-10-17

**Soundness:** 3
**Presentation:** 3
**Contribution:** 2
**Rating:** 2
**Confidence:** 4

**Summary:**

The paper proposes Meta-Memory for Memory Management (M^4), a four-stage framework consisting of category-based memory learning, chain-based construction, self-monitoring updates, and self-reflection retrieval. M^4 aims to let LLMs autonomously determine what to store, how to structure it, when to revise it, and how to retrieve it in long-term conversations. Reported experiments on two long-term dialog datasets and a long-term QA benchmark show improvements over strong baselines and noted reduced memory storage (over 50%) when using compression.

**Strengths:**

1. This paper proposed useful abstractions like category and memory unit. These abstractions can be helpful in designing storage primitive and indexing schemes in future systems.
1. M^4 has a lot of practical design choices, as exemplified by the numerous steps included in M^4. These designs are useful for practitioners designing real-world systems.
1. Extensive experiments are performed on multiple models, benchmarks, and baselines to confirm the usefulness of the system.

**Weaknesses:**

1. Besides the novel schema design, the overall technical novelty of the paper is below the acceptance threshold. It seems that M^4 consists of a large number of moving parts combining hand-crafted prompts with existing off-the-shelf modules like dense embeddings and prompt compressors. The scaffolding itself is also a set of heavily manually designed heuristics.
1. The intensive human engineering also limits the applicability of the memory system to user chat memory. By contrast, more general memory architectures can be applied to and are actually tested on long-context and RAG tasks. [1] [2]
1. Lacking cost analysis. The complicated prompting steps may lead to an unacceptable cost.  As this paper builds a real-world system, it is crucial that both latency and token cost is analyzed to justify the mild performance gain over the “context” baseline.

[1] From RAG to Memory: Non-Parametric Continual Learning for Large Language Models. Gutiérrez et al., 2025.

[2] M+: Extending memoryllm with scalable long-term memory. Wang et al., 2025.

**Questions:**

Please refer to the weaknesses.

---

### Official Review · Reviewer_Cww3 · 2025-10-20

**Soundness:** 3
**Presentation:** 3
**Contribution:** 3
**Rating:** 2
**Confidence:** 5

**Summary:**

The paper proposes M4, a meta-memory framework for long-term dialogue with LLMs. M4 adds four components: category-based memory learning to decide what to store, chain-based memory construction to order memories by category over time, self-monitoring updates with "calibrate" (resolve conflicts) and "compress" (LLMLingua) to shrink unused items, and self-reflection retrieval that traverses memory chains bidirectionally until the model is confident.

**Strengths:**

The paper targets a real pain point: deciding what to remember and how to use it during long, multi-session chats. The decomposition into learn/construct/update/retrieve is clear and tied to concrete prompts and an algorithm for retrieval with confidence checks; Algorithm 1 and prompt templates in the appendix improve clarity and reproducibility.

**Weaknesses:**

1. The model coverage is out of date for a 2025–2026 submission. The paper evaluates only Qwen2.5-7B, Llama-3-8B, GPT-3.5-Turbo-0125, and GPT-4o (2024-08-06). It does not include newer families such as GPT-5 or Qwen-3, and it does not study how gains scale with model size. The method also relies on a single embedding model (text-embedding-3-large) without analyzing how the choice of embedding affects retrieval and memory linking. The paper should add a broader sweep across current models and sizes, and report any interaction between backbone size, embedding choice, and the memory pipeline.

2. The experiments appear to be single-seed. There is no report of running multiple trials or reporting variance or confidence intervals for automatic metrics, G-Eval, or the human study.

3. The related work section omits closely related dynamic memory controllers. In particular, it does not cite or compare against From Isolated Conversations to Hierarchical Schemas: Dynamic Tree Memory Representation for LLMs [1], which also proposes dynamic memory saving and retrieval.

[1] Rezazadeh, Alireza, et al. "From isolated conversations to hierarchical schemas: Dynamic tree memory representation for llms." arXiv preprint arXiv:2410.14052 (2024).

**Questions:**

1. Model coverage and scaling. Will you add results with newer backbones (for example GPT-5, Qwen-3) and show how performance changes with model size on the same prompts and datasets? Please include a size sweep (small, medium, large) and report whether any module of your pipeline benefits more from larger models.

2. Embedding choice. Your pipeline relies on a single embedding model. Can you provide a study across several embedding models and dimensions, and report how retrieval accuracy, memory linking, and end metrics change? Please include ablations where you vary the embedding model while holding the backbone fixed.

3. Multiple runs and uncertainty. Many components are stochastic. Can you re-run all main results with several random seeds and report mean ± standard deviation, plus statistical tests? Please include variance for G-Eval and for the human study.

4. Human evaluation details. How many annotators did you use, how were they trained, what was inter-annotator agreement, and how were ties handled? Please add confidence intervals or bootstrap intervals for win rates.

---

### Official Review · Reviewer_RTgj · 2025-10-23

**Soundness:** 1
**Presentation:** 1
**Contribution:** 2
**Rating:** 2
**Confidence:** 4

**Summary:**

The paper proposes Meta-Memory for Memory Management  (M^4), a framework that improves the performance of Large Language Models (LLMs) on long-term conversations. It allows an explicit memory management of LLMs covering acquisition, construction, update, and retrieval. The method is validated by augmenting open-weights (Qwen2.5-7B, Llama3-8B) and closed LLMs (GPT-4o) and evaluating them on long-term conversation datasets. It is compared with multiple baselines.

**Strengths:**

- The method is validated both on open-weights and closed models.
- The summary-based baselines help situate the method's contribution.

**Weaknesses:**

- The description of the method is not very rigorous. A lot of important details are missing to properly understand let alone replicate the paper. The absence of accompanying code doesn't help either. The notations are not clear and not always properly defined. For example, in line 159 the method operates on "categories". What is a category? What does n0 in line 160 represent? In line 168, it is said that "M4 summarizes"  but how is this done exactly? What does the symbol in equation 3 mean?
- There are no details about how the human evaluation was carried out.
- The readability could be improved. The paper introduces concepts without providing references or concrete explanations
The font used in the figures are too small (Figure 2 and Figure 6). Figure 6 is hard to interpret. The corresponding caption does not clearly explain the different components of the figure and how they should be read. The unit of the y-axis in figure 4 is provided neither in the figure nor the caption. We cannot properly see the scores in Figure 2 a) and b)
- It is claimed in 4.3 that the method improves LLM's memory management and response quality. However, there is no qualitative analysis or examples of model generations even in the appendix.
- Important baselines that are not summary-based were not considered in the paper.

**Questions:**

- What exactly is a category?
- How was the human evaluation performed?
- How does the method compare to non summary-based baselines?

---

### Official Review · Reviewer_jFx8 · 2025-10-28

**Soundness:** 3
**Presentation:** 3
**Contribution:** 2
**Rating:** 4
**Confidence:** 4

**Summary:**

The paper proposes Meta-Memory for Memory Management, a framework that equips LLMs with self-monitoring and self-reflective mechanisms to manage long-term conversational memory. Concretely, it learns which information is worth memorizing via category-based memory learning, organizes it into chain-based structures for retrieval, updates memory with two actions—calibrate and compress, and performs self-reflection-based dynamic retrieval at inference time.

**Strengths:**

(1) Tackles a real, persistent gap in long-horizon conversational agents—what to store, how to maintain it, and how to retrieve it.

(2) Modular memory design (Learn → Construct → Update → Retrieve) sounds feasible, enabling structured storage (chains with cross-links), conflict resolution (calibrate/merge), and token-aware retention (compress) while keeping retrieval adaptive via a self-reflection policy.

**Weaknesses:**

(1) I think the method is an effective packaging of familiar primitives more than a new algorithmic contribution.

(2) G-Eval uses GPT-4o as judge while GPT-4o is also one of the evaluated models. There’s no statistical testing reported to temper evaluator bias.

(3) Core steps depend on prompts and LLM judgments (categorying, calibrate/merge, compression stop, retrieval keep/skip/stop). There’s no robustness study over temperatures.

(4) A brief taxonomy (wrong merges during calibrate, over-compression, stale/irrelevant chains, premature stopping) with examples would be more helpful.

**Questions:**

(1) If you’re using G-Eval, can you repeat with a different LLM judge and report significance tests on the main metrics to reduce coupling with GPT-4o?

(2) Could you share a small sweep over temperatures for categorying, calibrate/merge, compression stop, and retrieval decisions?

(3) Please add a short error taxonomy (3–5 concrete examples) and note which module would mitigate each case.

(4) A compact table with per-dialogue token counts and $ split by module (learn/construct/update/retrieve) would make deployment trade-offs clearer.

---

### Note · Authors · 2025-11-25

I have read and agree with the venue's withdrawal policy on behalf of myself and my co-authors.